# Sampled Softmax with Random Fourier Features

**Ankit Singh Rawat, Jiecao Chen, Felix Yu, Ananda Theertha Suresh, and Sanjiv Kumar**
Google Research, New York
{ankitsrawat, chenjiecao, felixyu, theertha, sanjivk}@google.com

## Abstract

The computational cost of training with softmax cross entropy loss grows linearly with the number of classes. For the settings where a large number of classes are involved, a common method to speed up training is to sample a subset of classes and utilize an estimate of the loss gradient based on these classes, known as the *sampled softmax* method. However, the sampled softmax provides a biased estimate of the gradient unless the samples are drawn from the exact softmax distribution, which is again expensive to compute. Therefore, a widely employed practical approach involves sampling from a simpler distribution in the hope of approximating the exact softmax distribution. In this paper, we develop the first theoretical understanding of the role that different sampling distributions play in determining the quality of sampled softmax. Motivated by our analysis and the work on kernel-based sampling, we propose the *Random Fourier Softmax* (RF-softmax) method that utilizes the powerful Random Fourier Features to enable more efficient and accurate sampling from an approximate softmax distribution. We show that RF-softmax leads to low bias in estimation in terms of both the full softmax distribution and the full softmax gradient. Furthermore, the cost of RF-softmax scales only logarithmically with the number of classes.

## 1 Introduction

The cross entropy loss based on softmax function is widely used in multi-class classification tasks such as natural language processing [1], image classification [2], and recommendation systems [3]. In multi-class classification, given an input $\mathbf{x} \in \mathcal{X}$, the goal is to predict its class $t \in \{1, 2, \ldots, n\}$, where $n$ is the number of classes. Given an input feature $\mathbf{x}$, the model (often a neural network) first computes an input embedding $\mathbf{h} \in \mathbb{R}^d$ and then the raw scores or *logits* for classes $\mathbf{o} = (o_1, \ldots, o_n)$ as the product of the input embedding $\mathbf{h}$ and the class embeddings $\mathbf{c}_1, \ldots, \mathbf{c}_n \in \mathbb{R}^d$,

$$o_i = \tau \mathbf{h}^T \mathbf{c}_i. \tag{1}$$

Here, $\tau$ is often referred to as the (inverse) *temperature* parameter of softmax. Given the logits, the probability that the model assigns to the $i$-th class is computed using the *full softmax* function

$$p_i = e^{o_i}/Z, \tag{2}$$

where $Z = \sum_{i=1}^{n} e^{o_i}$ is called the *partition function*. The distribution in (2) is commonly referred to as the softmax distribution. Given a training set, the model parameters are estimated by minimizing an empirical risk over the training set, where the empirical risk is defined by the *cross-entropy loss* based on softmax function or the *full softmax loss*. Let $t \in [n]$ denote the true class for the input $\mathbf{x}$, then the full softmax loss is defined as[1]

$$\mathcal{L}(\mathbf{x}, t) := -\log p_t = -o_t + \log Z. \tag{3}$$

One typically employs first order optimization methods to train neural network models. This requires computing the gradient of the loss with respect to the model parameter $\boldsymbol{\theta}$ during each iteration

$$\nabla_{\boldsymbol{\theta}}\mathcal{L}(\mathbf{x}, t) = -\nabla_{\boldsymbol{\theta}} o_t + \sum_{i=1}^{n} \frac{e^{o_i}}{Z} \cdot \nabla_{\boldsymbol{\theta}} o_i = -\nabla_{\boldsymbol{\theta}} o_t + \mathbb{E}_{s \sim p}\left[\nabla_{\boldsymbol{\theta}} o_s\right], \tag{4}$$

where the expectation is taken over the softmax distribution (cf. (2)). As evident from (4), computing the gradient of the full softmax loss takes $\mathcal{O}(dn)$ time due to the contributions from all $n$ classes. Therefore, training a model using the full softmax loss becomes prohibitively expensive in the settings where a large number of classes are involved. To this end, various approaches have been proposed for efficient training. This includes different modified loss functions: hierarchical softmax [5] partitions the classes into a tree based on class similarities, allowing for $\mathcal{O}(d \log n)$ training and inference time; spherical softmax [6, 7] replaces the exponential function by a quadratic function, enabling efficient algorithm to compute the updates of the output weights irrespective of the output size. Efficient hardware-specific implementations of softmax are also being actively studied [8].

## 1.1 Sampled softmax

A popular approach to speed up the training of full softmax loss is using *sampled softmax*: instead of including all classes during each iteration, a small random subset of $n$ classes is considered, where each *negative* class is sampled with some probability. Formally, let the number of sampled classes during each iteration be $m$, with class $i$ being picked with probability $q_i$. Let $\mathcal{N}_t \triangleq [n] \backslash \{t\}$ be the set of negative classes. Assuming that $s_1, \ldots, s_m \in \mathcal{N}_t$ denote the sampled class indices, following [9], we define the adjusted logits $\mathbf{o}' = \{o'_1, o'_2, \ldots, o'_{m+1}\}$ such that $o'_1 = o_t$ and for $i \in [m]$,

$$o'_{i+1} = o_{s_i} - \log(m q_{s_i}). \tag{5}$$

Accordingly, we define the *sampled softmax distribution* as $p'_i = \frac{e^{o'_i}}{Z'}$, where $Z' = \sum_{j=1}^{m+1} e^{o'_j}$. The *sampled softmax loss* corresponds to the cross entropy loss with respect to the sampled softmax distribution:

$$\mathcal{L}'(\mathbf{x}, t) = -\log p'_t = -o_t + \log Z'. \tag{6}$$

Here, we note that adjusting the logits for the sampled negative classes using their expected number of occurrence in (5) ensures that $Z'$ is an unbiased estimator of $Z$ [9]. Since $\mathcal{L}'(\mathbf{x}, t)$ depends only on $m + 1$ classes, the computational cost is reduced from $\mathcal{O}(dn)$ to $\mathcal{O}(dm)$ as compared to the full softmax loss in (3).

In order to realize the training with the full softmax loss, one would like the gradient of the sampled softmax loss to be an unbiased estimator of the gradient of the full softmax loss[2], i.e.,

$$\mathbb{E}\left[\nabla_{\boldsymbol{\theta}} \mathcal{L}'\right] = \nabla_{\boldsymbol{\theta}} \mathcal{L}, \tag{7}$$

where the expectation is taken over the sampling distribution $q$. As it turns out, the sampling distribution plays a crucial role in ensuring the unbiasedness of $\nabla_{\boldsymbol{\theta}} \mathcal{L}'$. Bengio and Senécal [9] show that (7) holds if the sampling distribution is the full softmax distribution itself, i.e., $q_i \propto e^{o_i}$ (cf. (2)).

However, sampling from the softmax distribution itself is again computationally expensive: one needs to compute the partition function $Z$ during each iteration, which is again an $\mathcal{O}(dn)$ operation since $Z$ depends both on the current model parameters and the input. As a feasible alternative, one usually samples from a distribution which does not depend on the current model parameters and the input. Common choices are uniform, log-uniform, or the global prior of classes [10, 1]. However, since these distributions are far from the full softmax distribution, they can lead to significantly worse solutions. Various approaches have been proposed to improve negative sampling. For example, a separate model can be used to track the distribution of softmax in language modeling tasks [9]. One can also use an LSH algorithm to find the approximate nearest classes in the embedding space which in turn helps in sampling from the softmax distribution efficiently [11]. Quadratic kernel softmax [12] uses a kernel-based sampling method and quadratic approximation of the softmax function to draw each sample in sublinear time. Similarly, the Gumbel trick has been proposed to sample from the softmax distribution in sublinear time [13]. The partition function can also be written in a double-sum formulation to enable an unbiased sampling algorithm for SGD [14, 15].

Among other training approaches based on sampled losses, Noise Contrastive Estimation (NCE) and its variants avoid computing the partition function [16], and (semi-)hard negative sampling [17, 4, 18] selects the negatives that most violate the current objective function. Hyvärinen [19] proposes minimization of Fisher divergence (a.k.a. score matching) to avoid computation of the partition function $Z$. However, in our setting, the partition function depends on the input embedding $\mathbf{h}$, which changes during the training. Thus, while calculating the score function (taking derivative of $Z$ with respect to $(\mathbf{h}, \mathbf{c})$), the partition function has a non-trivial contribution which makes this approach inapplicable to our setting. We also note the existence of MCMC based approaches in the literature (see, e.g., [20]) for sampling classes with a distribution that is close to the softmax distribution. Such methods do not come with precise computational complexity guarantees.

## 1.2 Our contributions

**Theory.** Despite a large body of work on improving the quality of sampled softmax, developing a theoretical understanding of the performance of sampled softmax has not received much attention. Blanc and Rendle [12] show that the full softmax distribution is the *only* distribution that provides an unbiased estimate of the true gradient $\nabla_{\boldsymbol{\theta}} \mathcal{L}$. However, it is not clear *how different sampling distributions affect the bias* $\nabla_{\boldsymbol{\theta}} \mathcal{L} - \mathbb{E}\left[\nabla_{\boldsymbol{\theta}} \mathcal{L}'\right]$. In this paper, we address this issue and characterize the bias of the gradient for a generic sampling distribution (cf. Section 2).

**Algorithm.** In Section 3, guided by our analysis and recognizing the practical appeal of kernel-based sampling [12], we propose Random Fourier softmax (RF-softmax), a new kernel-based sampling method for the settings with normalized embeddings. RF-softmax employs the powerful Random Fourier Features [21] and guarantees small bias of the gradient estimate. Furthermore, the complexity of sampling one class for RF-softmax is $\mathcal{O}(D \log n)$, where $D$ denotes the number of random features used in RF-softmax. In contrast, assuming that $d$ denotes the embedding dimension, the full softmax and the prior kernel-based sampling method (Quadratic-softmax [12]) incur $\mathcal{O}(dn)$ and $\mathcal{O}(d^2 \log n)$ computational cost to generate one sample, respectively. In practice, $D$ can be two orders of magnitudes smaller than $d^2$ to achieve similar or better performance. As a result, RF-softmax has two desirable features: 1) better accuracy due to lower bias and 2) computational efficiency due to low sampling cost.

**Experiments.** We conduct experiments on widely used NLP and extreme classification datasets to demonstrate the utility of the proposed RF-softmax method (cf. Section 4).

## 2 Gradient bias of sampled softmax

The goal of sampled softmax is to obtain a computationally efficient estimate of the *true gradient* $\nabla_{\boldsymbol{\theta}} \mathcal{L}$ (cf. (4)) of the full softmax loss (cf. (3)) with small bias. In this section we develop a theoretical understanding of how different sampling distributions affect the bias of the gradient. To the best of our knowledge, this is the first result of this kind.

For the cross entropy loss based on the sampled softmax (cf. (6)), the training algorithm employs the following estimate of $\nabla_{\boldsymbol{\theta}} \mathcal{L}$.

$$\nabla_{\boldsymbol{\theta}} \mathcal{L}' = -\nabla_{\boldsymbol{\theta}} o_t + \frac{e^{o_t} \nabla_{\boldsymbol{\theta}} o_t + \sum_{i \in [m]} \frac{e^{o_{s_i}}}{m q_{s_i}} \nabla_{\boldsymbol{\theta}} o_{s_i}}{e^{o_t} + \sum_{i \in [m]} \frac{e^{o_{s_i}}}{m q_{s_i}}}. \tag{8}$$

The following result bounds the bias of the estimate $\nabla_{\boldsymbol{\theta}} \mathcal{L}'$. Without loss of generality, we work with the sampling distributions that assign strictly positive probability to each negative class, i.e., $q_i > 0 \ \forall \ i \in \mathcal{N}_t$.

**Theorem 1.** *Let $\nabla_{\boldsymbol{\theta}} \mathcal{L}'$ (cf. (8)) be the estimate of $\nabla_{\boldsymbol{\theta}} \mathcal{L}$ based on $m$ negative classes $s_1, \ldots, s_m$, drawn according to the sampling distribution $q$. We further assume that the gradients of the logits $\nabla_{\boldsymbol{\theta}} o_i$ have their coordinates bounded[3] by $M$. Then, the bias of $\nabla_{\boldsymbol{\theta}} \mathcal{L}'$ satisfies*

$$\mathrm{LB} \leq \mathbb{E}\left[\nabla_{\boldsymbol{\theta}} \mathcal{L}'\right] - \nabla_{\boldsymbol{\theta}} \mathcal{L} \leq \mathrm{UB} \tag{9}$$

*with*

$$\text{LB} \triangleq -\frac{M \sum_{k \in \mathcal{N}_t} e^{o_k} \left| Z_t - \frac{e^{o_k}}{q_k} \right|}{mZ^2} \left( 1 - o\left(\frac{1}{m}\right) \right) \cdot \mathbf{1}, \tag{10}$$

$$\text{UB} \triangleq \left( \underbrace{\frac{\sum_{j \in \mathcal{N}_t} \frac{e^{2o_j}}{q_j} - Z_t^2}{mZ^3} + o\left(\frac{1}{m}\right)}_{\text{UB}_1} \right) \cdot \mathbf{g} + \left( \frac{2M}{m} \underbrace{\frac{\max_{i,i' \in \mathcal{N}_t} \left| \frac{e^{o_i}}{q_i} - \frac{e^{o_{i'}}}{q_{i'}} \right| Z_t}{Z^2 + \sum_{j \in \mathcal{N}_t} \frac{e^{2o_j}}{q_j}}}_{\text{UB}_2} + o\left(\frac{1}{m}\right) \right) \cdot \mathbf{1}, \tag{11}$$

*where $Z_t \triangleq \sum_{j \in \mathcal{N}_t} e^{o_j}$, $\mathbf{g} \triangleq \sum_{j \in \mathcal{N}_t} e^{o_j} \nabla_{\boldsymbol{\theta}} o_j$ and $\mathbf{1}$ is the all one vector.*

The proof of Theorem 1 is presented in Appendix A. Theorem 1 captures the effect of the underlying sampling distribution $q$ on the bias of gradient estimate in terms of three (closely related) quantities:

$$\sum_{j \in \mathcal{N}_t} \frac{e^{2o_j}}{q_j}, \quad \max_{j,j' \in \mathcal{N}_t} \left| \frac{e^{o_j}}{q_j} - \frac{e^{o_{j'}}}{q_{j'}} \right|, \text{ and } \left| \sum_{j \in \mathcal{N}_t} e^{o_j} - \frac{e^{o_k}}{q_k} \right|. \tag{12}$$

Ideally, we would like to pick a sampling distribution for which all these quantities are as small as possible. Since $q$ is a probability distribution, it follows from Cauchy-Schwarz inequality that

$$\sum_j \frac{e^{2o_j}}{q_j} = \left( \sum_j q_j \right) \cdot \left( \sum_j \frac{e^{2o_j}}{q_j} \right) \geq \left( \sum_j e^{o_j} \right)^2. \tag{13}$$

If $q_j \propto e^{o_j}$, then (13) is attained (equivalently, $\sum_j \frac{e^{2o_j}}{q_j}$ is minimized). In particular, this implies that UB$_1$ in (11) disappears. Furthermore, for such a distribution we have $q_j = \frac{e^{o_j}}{\sum_{i \in \mathcal{N}_t} e^{o_i}}$. This implies that both UB$_2$ and LB disappear for such a distribution as well. This guarantees a small bias of the gradient estimate $\nabla_{\boldsymbol{\theta}} \mathcal{L}'$.

Since sampling exactly from the distribution $q$ such that $q_j \propto e^{o_j}$ is computationally expensive, one has to resort to other distributions that incur smaller computational cost. However, to ensure small bias of the gradient estimate $\nabla_{\boldsymbol{\theta}} \mathcal{L}'$, Theorem 1 and the accompanying discussion suggest that it is desirable to employ a distribution that ensures that $\frac{e^{o_j}}{q_j}$ is as close to 1 as possible for each $j \in \mathcal{N}_t$ and all possible values of the logit $o_j$. In other words, we are interested in those sampling distributions that provide a tight *uniform multiplicative approximation* of $e^{o_j}$ in a computationally efficient manner.

This motivates our main contribution in the next section, where we rely on kernel-based sampling methods to efficiently implement a distribution that uniformly approximates the softmax distribution.

## 3 Random Fourier Softmax (RF-Softmax)

In this section, guided by the conclusion in Section 2, we propose Random Fourier Softmax (RF-softmax), as a new sampling method that employs Random Fourier Features to tightly approximate the full softmax distribution. RF-softmax falls under the broader class of kernel-based sampling methods which are amenable to efficient implementation. Before presenting RF-softmax, we briefly describe the kernel-based sampling and an existing method based on quadratic kernels [12].

### 3.1 Kernel-based sampling and Quadratic-softmax

Given a kernel $K : \mathbb{R}^d \times \mathbb{R}^d \to \mathbb{R}$, the input embedding $\mathbf{h} \in \mathbb{R}^d$, and the class embeddings $\mathbf{c}_1, \ldots, \mathbf{c}_n \in \mathbb{R}^d$, kernel-based sampling selects the class $i$ with probability $q_i = \frac{K(\mathbf{h}, \mathbf{c}_i)}{\sum_{j=1}^n K(\mathbf{h}, \mathbf{c}_j)}$. Note that if $K(\mathbf{h}, \mathbf{c}_i) = \exp(o_i) = \exp(\tau \mathbf{h}^T \mathbf{c}_i)$, this amounts to directly sampling from the softmax distribution.

Blanc and Steffen [12] show that if the kernel can be linearized by a mapping $\phi : \mathbb{R}^d \to \mathbb{R}^D$ such that $K(\mathbf{h}, \mathbf{c}_i) = \phi(\mathbf{h})^T \phi(\mathbf{c}_i)$, sampling one point from the distribution takes only $\mathcal{O}(D \log n)$ time by a divide-and-conquer algorithm. We briefly review the algorithm in this section.

Under the linearization assumption, the sampling distribution takes the following form.

$$q_i = \frac{K(\mathbf{h}, \mathbf{c}_i)}{\sum_{j=1}^n \phi(\mathbf{h})^T \phi(\mathbf{c}_j)} = \frac{\phi(\mathbf{h})^T \phi(\mathbf{c}_i)}{\phi(\mathbf{h})^T \sum_{j=1}^n \phi(\mathbf{c}_j)}.$$

The idea is to organize the classes in a binary tree with individual classes at the leaves. We then sample along a path on this tree recursively until we reach a single class. Each sampling step takes $\mathcal{O}(D)$ time as we can pre-compute $\sum_{j \in S} \phi(\mathbf{c}_j)$ where $S$ is any subset of classes. Similarly, when the embedding of a class changes, the cost of updating all $\sum_{j \in S} \phi(\mathbf{c}_j)$ along the path between the root and this class is again $\mathcal{O}(D \log n)$.

Note that we pre-compute $\sum_{j \in [n]} \phi(\mathbf{c}_j)$ for the root node. Now, suppose the left neighbor and the right neighbor of the root node divide all the classes into two disjoint set $S_1$ and $S_2$, respectively. In this case, we pre-compute $\sum_{j \in S_1} \phi(\mathbf{c}_j)$ and $\sum_{j \in S_2} \phi(\mathbf{c}_j)$ for the left neighbor and the right neighbor of the root node, respectively. First, the probability of the sampled class being in $S_1$ is

$$q_{S_1} = \frac{\phi(\mathbf{h})^T \sum_{j \in S_1} \phi(\mathbf{c}_j)}{\phi(\mathbf{h})^T \sum_{j \in S_1} \phi(\mathbf{c}_j) + \phi(\mathbf{h})^T \sum_{j \in S_2} \phi(\mathbf{c}_j)} \tag{14}$$

And the probability of the sampled class being in $S_2$ is $1 - q_{S_1}$. We then recursively sample along a path until a single classes is reached, i.e., we reach a leaf node. After updating the embedding of the sampled class, we recursively trace up the tree to update the sums stored on all the nodes until we reach the root node.

Given the efficiency of the kernel-based sampling, Blanc and Steffen [12] propose a sampled softmax approach which utilizes the quadratic kernel to define the sampling distribution.

$$K_{\text{quad}}(\mathbf{h}, \mathbf{c}_i) = \alpha \cdot (\mathbf{h}^T \mathbf{c}_i)^2 + 1. \tag{15}$$

Note that the quadratic kernel can be explicitly linearized by the mapping $\phi(\mathbf{z}) = [\sqrt{\alpha} \cdot (\mathbf{z} \otimes \mathbf{z}), 1]$. This implies that $D = \mathcal{O}(d^2)$; consequently, sampling one class takes $\mathcal{O}(d^2 \log n)$ time. Despite the promising results of Quadratic-softmax, it has the following caveats:

- The quadratic kernel with $\alpha = 100$, the value used in [12], does not give a tight multiplicative approximation of the exponential kernel $e^{o_j}$. Thus, according to the analysis in Section 2, this results in a gradient estimate with large bias.

- The $\mathcal{O}(d^2)$ computational cost can be prohibitive for models with large embedding dimensions.

- Since a quadratic function is a poor approximation of the exponential function for negative numbers, it is used to approximate a modified *absolute softmax* loss function in [12] instead, where *absolute* values of logits serve as input to the softmax function.

Next, we present a novel kernel-based sampling method that addresses all of these shortcomings.

## 3.2   RF-softmax

Given the analysis in Section 2 and low computational cost of the linearized kernel-based sampling methods, our goal is to come up with a linearizable kernel (better than quadratic) that provides a good uniform multiplicative approximation of the exponential kernel $K(\mathbf{h}, \mathbf{c}) = e^o = e^{\tau \mathbf{h}^T \mathbf{c}}$. More concretely, we would like to find a nonlinear map $\phi(\cdot) : \mathbb{R}^d \to \mathbb{R}^D$ such that the error between $K(\mathbf{h}, \mathbf{c})$ and $\hat{K}(\mathbf{h}, \mathbf{c}) = \phi(\mathbf{h})^T \phi(\mathbf{c})$ is small for all values of $\mathbf{h}$ and $\mathbf{c}$.

The Random Maclaurin Features [22] seem to be an obvious choice here since it provides an unbiased estimator of the exponential kernel. However, due to rank deficiency of the produced features, it requires large $D$ in order to achieve small mean squared error [23, 24]. We verify that this is indeed a poor choice in Table 1.

In contrast, the Random Fourier Features (RFF) [21] is much more compact [23]. Moreover, these features and their extensions are also theoretically better understood (see, e.g., [25, 26, 27]). This leads to the natural question: *Can we use RFF to approximate the exponential kernel?* However, this approach faces a major challenge at the outset: RFF only works for positive definite shift-invariant kernels such as the Gaussian kernel, while the exponential kernel is not shift-invariant.

A key observation is that when the input embedding $\mathbf{h}$ and the class embedding $\mathbf{c}$ are normalized, the exponential kernel becomes the Gaussian kernel (up to a multiplicative constant):

$$e^{\tau \mathbf{h}^T \mathbf{c}} = e^\tau e^{-\frac{\tau ||\mathbf{h} - \mathbf{c}||_2^2}{2}}. \tag{16}$$

| Method | Quadratic [12] | Random Fourier [21] | | | Random Maclaurin [22] |
|---|---|---|---|---|---|
| $D$ | $256^2$ | 100 | 1000 | $256^2$ | $256^2$ |
| MSE | 2.8e-3 | 2.6e-3 | 2.7e-4 | 5.5e-6 | 8.8e-2 |

Table 1: Mean squared error (MSE) of approximating a kernel $\exp(\tau \mathbf{h}^T \mathbf{c})$. $\mathbf{h}$ and $\mathbf{c}$ are randomly sampled from the USPS dataset ($d = 256$). The data is normalized, i.e., $||\mathbf{h}||_2 = 1$, $||\mathbf{c}||_2 = 1$. For Quadratic, we assume the form is $\alpha \cdot (\mathbf{h}^T \mathbf{c}_i)^2 + \beta$ and solve $\alpha$ and $\beta$ in a linear system to get the optimal MSE. In practice, with fixed $\alpha$ and $\beta$ as in [12], the MSE will be larger. Random Fourier has much lower MSE with same $D$, and much smaller $D$ with similar MSE. Also note that Random Fourier and Random Maclaurin are unbiased.

We note that normalized embeddings are widely used in practice to improve training stability and model quality [28, 29, 30]. In particular, it attains improved performance as long as $\tau$ (cf (1)) is large enough to ensure that the output of softmax can cover (almost) the entire range (0,1). In Section 4, we verify that restricting ourselves to the normalized embeddings does not hurt and, in fact, improves the final performance.

For the Gaussian kernel $K(\mathbf{x} - \mathbf{y}) = e^{-\frac{\nu \|\mathbf{x} - \mathbf{y}\|^2}{2}}$ with temperature parameter $\nu$, the $D$-dimensional RFF map takes the following form.

$$\phi(\mathbf{u}) = \frac{1}{\sqrt{D}} \Big[ \cos(\mathbf{w}_1^T \mathbf{u}), \dots, \cos(\mathbf{w}_D^T \mathbf{u}), \sin(\mathbf{w}_1^T \mathbf{u}), \dots, \sin(\mathbf{w}_D^T \mathbf{u}) \Big], \tag{17}$$

where $\mathbf{w}_1, \dots, \mathbf{w}_D \sim N(0, \mathbf{I}/\nu)$. The RFF map provides an unbiased approximation of the Gaussian kernel [26, Lemma 1]

$$e^{-\frac{\nu \|\mathbf{x} - \mathbf{y}\|^2}{2}} \approx \phi(\mathbf{x})^T \phi(\mathbf{y}). \tag{18}$$

Now, given an input embedding $\mathbf{h}$, if we sample class $i$ with probability $q_i \propto \exp\left(-\tau \|\mathbf{c}_i - \mathbf{h}\|^2/2\right)$, then it follows from (16) that our sampling distribution is the same as the softmax distribution. Therefore, with normalized embeddings, one can employ the kernel-based sampling to realize the sampled softmax such that class $i$ is sampled with the probability

$$q_i \propto \phi(\mathbf{c}_i)^T \phi(\mathbf{h}). \tag{19}$$

We refer to this method as Random Fourier Softmax (RF-softmax). RF-softmax costs $\mathcal{O}(D \log n)$ to sample one point (cf. Section 3.1). Note that computing the nonlinear map takes $\mathcal{O}(Dd)$ time with the classic Random Fourier Feature. One can easily use the structured orthogonal random feature (SORF) technique [26] to reduce this complexity to $\mathcal{O}(D \log d)$ with even lower approximation error. Since typically the embedding dimension $d$ is on the order of hundreds and we consider large $n$: $d \ll n$, the overall complexity of RF-softmax is $\mathcal{O}(D \log n)$.

As shown in Table 1, the RFF map approximates the exponential kernel with the mean squared error that is orders of magnitudes smaller than that for the quadratic map with the same mapping dimension $D$. The use of RFF raises interesting implementation related challenges in terms of selecting the temperature parameter $\nu$ to realize low biased sampling, which we address in the following subsection.

### 3.3 Analysis and discussions

Recall that the discussion following Theorem 1 implies that, for low-bias gradient estimates, the sampling distribution $q_i$ needs to form a tight multiplicative approximation of the softmax distribution $p_i$, where $p_i \propto \exp(o_i) = \exp(\tau \mathbf{h}^T \mathbf{c}_i)$. In the following result, we quantify the quality of the proposed RF-softmax based on the ratio $|p_i/q_i|$. The proof of the result is presented in Appendix B.

**Theorem 2.** *Given the $\ell_2$-normalized input embedding $\mathbf{h}$ and $\ell_2$-normalized class embeddings $\{\mathbf{c}_1, \dots, \mathbf{c}_n\}$, let $o_i = \tau \mathbf{h}^T \mathbf{c}_i$ be the logits associated with the class $i$. Let $q_i$ denote the probability of sampling the $i$-th class based on $D$-dimensional Random Fourier Features, i.e., $q_i = \frac{1}{C} \cdot \phi(\mathbf{c}_i)^T \phi(\mathbf{h})$, where $C$ is the normalizing constant. Then, as long as $e^{2\nu} \leq \frac{\gamma}{\rho \sqrt{d}} \cdot \frac{\sqrt{D}}{\log D}$, the following holds with probability at least $1 - \mathcal{O}\left(\frac{1}{D^2}\right)$.*

$$e^{(\tau - \nu)\mathbf{h}^T \mathbf{c}_i} \cdot (1 - 2\gamma) \leq \frac{1}{\sum_{i \in \mathcal{N}_t} e^{o_i}} \cdot \left| \frac{e^{o_i}}{q_i} \right| \leq e^{(\tau - \nu)\mathbf{h}^T \mathbf{c}_i} \cdot (1 + 4\gamma), \tag{20}$$

*where $\gamma$ and $\rho$ are positive constants.*

**Remark 1.** *With large enough $D$, we may invoke Theorem 2 with $\nu = \tau$ and*

$$\gamma = a' \cdot \left( e^{2\tau} \cdot \frac{\rho \sqrt{d} \log D}{\sqrt{D}} \right),$$

*where $a' > 1$ is a fixed constant. In this case, since $\gamma = o_D(1)$, it follows from (20) that $q_i \propto (1 \pm o_D(1)) \cdot p_i$, for each $i \in \mathcal{N}_t$. In particular, at $D = \infty$, we have $q_i \propto p_i$.*

We now combine Theorem 1 and Theorem 2 to obtain the following corollary, which characterizes the bias of the gradient estimate for RF-softmax in the regime where $D$ is large. The proof is presented in Appendix C.

**Corollary 1.** *Let $q_i = \frac{1}{C} \cdot \phi(\mathbf{c}_i)^T \phi(\mathbf{h})$ denote the probability of sampling the $i$-th class under RF-softmax. Then, with high probability, for large enough $D$, by selecting $\nu = \tau$ ensures that*

$$-o_D(1) \cdot \mathbf{1} \leq \mathbb{E}[\nabla_{\boldsymbol{\theta}} \mathcal{L}'] - \nabla_{\boldsymbol{\theta}} \mathcal{L} \leq \Big( o_D(1) + o\big(1/m\big) \Big) \cdot \mathbf{g} + \Big( o_D(1) + o\big(1/m\big) \Big) \cdot \mathbf{1},$$

*where $\mathbf{g} \triangleq \sum_{j \in \mathcal{N}_t} e^{o_j} \nabla_{\boldsymbol{\theta}} o_j$ and $\mathbf{1}$ is the all one vector. Here, the expectation is taken over the sampling distribution $q$.*

Note that Theorem 2 and Remark 1 highlight an important design issue in the implementation of the RF-softmax approach. The ability of $q$ to approximate $p$ (as stated in (20)) degrades as the difference $|\tau - \nu|$ increases. Therefore, one would like to pick $\nu$ to be as close to $\tau$ as possible (ideally exactly equal to $\tau$). However, for the fixed dimensional feature map $\phi$, the approximation guarantee in (20) holds for only those values of $\nu$ such that $e^{2\nu} \leq \frac{\gamma}{\rho \sqrt{d}} \cdot \frac{\sqrt{D}}{\log D}$. Therefore, the dimension of the feature map dictates which Gaussian kernels we can utilize in the proposed RF-softmax approach. On the other hand, the variance of the kernel approximation of Random Fourier feature grows with $\nu$ [26]. Additionally, in order to work with the normalized embeddings, it's necessary to select a reasonably large value for the temperature parameter[4] $\tau$. Therefore, choosing $\nu = \tau$ in this case will result in larger variance of the estimated kernel.

**Remark 2.** *As a trade off between bias and variance, while approximating the exponential kernel with a limited $D$ and large $\tau$, $\nu$ should be set as a value smaller than $\tau$.*

In Section 4, we explore different choices for the value of $\nu$ and confirm that some $\nu < \tau$ achieves the best empirical performance.

## 4   Experiments

In this section, we experimentally evaluate the proposed RF-softmax and compare it with several baselines using simple neural networks to validate the utility of the proposed sampling method and the accompanying theoretical analysis. We show that, in terms of the final model quality, RF-softmax with computational cost $\mathcal{O}(D \log n)$ ($D \ll d^2$) performs at par with the sampled softmax approach based on the full softmax distribution which incurs the computational cost of $\mathcal{O}(dn)$. We also show that RF-softmax outperforms Quadratic-softmax [12] that uses the quadratic function to approximate the exponential kernel and has computational cost $\mathcal{O}(d^2 \log n)$. In addition, we highlight the effect of different design choices regarding the underlying RFF map $\phi$ (cf. (17)) on the final performance of RF-softmax.

### 4.1   Datasets and models

**NLP datasets.** PENNTREEBANK [31] is a popular benchmark for NLP tasks with a vocabulary of size $10,000$. We train a language model using LSTM, where the normalized output of the LSTM serves as the input embedding. BNEWS [32] is another NLP dataset. For this dataset, we select the most frequent $64,000$ words as the vocabulary. Our model architecture for BNEWS is the same as the one used for PENNTREEBANK with more parameters. We fix the embedding dimension to be $d = 200$ for PENNTREEBANK and $d = 512$ for BNEWS.

**Extreme classification datasets**. We test the proposed method on three classification datasets with a large number of classes [33]. For each data point that is defined by a $v$-dimensional sparse feature vector, we first map the data point to a 128-dimensional vector by using a $v \times 128$ matrix. Once normalized, this vector serves as the input embedding $\mathbf{h}$. For $i \in [n]$, class $i$ is mapped to a 128-dimensional normalized vector $\mathbf{c}_i$. Following the convention of extreme classification literature [34, 35, 4], we report precision at $k$ (PREC@K) on the test set.

Recall that the proposed RF-softmax draws samples with computational complexity $\mathcal{O}(D \log n)$. We compare it with the following baselines.

- FULL, the full softmax loss, which has computational complexity $\mathcal{O}(dn)$.
- EXP, the sampled softmax approach with the full softmax distribution (cf. (2)) as the sampling distribution. This again has computational complexity $\mathcal{O}(dn)$ to draw one sample.
- UNIFORM, the sampled softmax approach that draws negative classes uniformly at random, amounting to $\mathcal{O}(1)$ computational complexity to draw one sample.
- QUADRATIC, the sampled softmax approach based on a quadratic kernel (cf. (15)). We follow the implementation of [12] and use $\alpha o^2 + 1$ with $\alpha = 100$. Since $\alpha o^2 + 1$ is a better approximation of $e^{|o|}$ than it is of $e^o$, [12] proposes to use the absolute softmax function $\tilde{p}_i = \frac{e^{\tau|o_i|}}{\sum_{j \in [n]} e^{\tau|o_j|}}$ while computing the cross entropy loss. We employ this modified loss function for the quadratic kernel as it gives better results as compared to the full softmax loss in (3). The cost of drawing one sample with this method is $\mathcal{O}(d^2 \log n)$.

Here, we refer to $1/\sqrt{\tau}$ as the temperature of softmax (cf. (2)). We set this parameter to $0.3$ as it leads to the best performance for the FULL baseline. This is a natural choice given that sampled softmax aims at approximating this loss function with a small subset of (sampled) negative classes.

| # classes ($n$) | Method | Wall time |
|---|---|---|
| 10,000 | EXP | 1.4 ms |
| | QUADRATIC | 6.5 ms |
| | RFF ($D = 50$) | 0.5 ms |
| | RFF ($D = 200$) | 0.6 ms |
| | RFF ($D = 500$) | 1.2 ms |
| | RFF ($D = 1,000$) | 1.4 ms |
| 500,000 | EXP | 32.3 ms |
| | QUADRATIC | 8.2 ms |
| | RFF ($D = 50$) | 1.6 ms |
| | RFF ($D = 200$) | 1.7 ms |
| | RFF ($D = 500$) | 2.0 ms |
| | RFF ($D = 1,000$) | 2.4 ms |

## 4.2 Experimental results

**Wall time.** We begin with verifying that RF-softmax indeed incurs low sampling cost. In Table 2, we compare the wall-time that different model dependent sampling methods take to compute the sampled softmax loss (cf. (6)) for different sets of parameters.

Table 2: Comparison of wall time for computing sampled softmax loss for different model-dependent sampling methods. (Batch size = 10, $m = 10$ and $d = 64$.)

**Normalized vs. unnormalized embeddings.** To justify our choice of working with normalized embeddings, we ran experiments on FULL with and without embedding normalization for both PENNTREEBANK and AMAZONCAT-13K. On PENNTREEBANK, after 10 epochs, the unnormalized version has much worse validation perplexity of 126 as compared to 120 with the normalized version. On AMAZONCAT-13K, both versions have PREC@1 87%.

Next, we discuss the key design choices for RF-softmax: the choice of the Gaussian sampling kernel defined by $\nu$ (cf. (16)); and the dimension of the RFF map $D$.

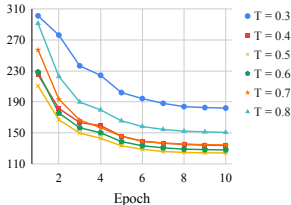

Figure 1: Validation perplexity for RF-softmax on PENNTREE-BANK with $m = 100$, $D = 1024$ and varying values of $T$.

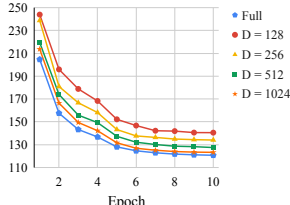

Figure 2: Validation perplexity for RF-softmax on PENNTREE-BANK with $m = 100$ and varying $D$.

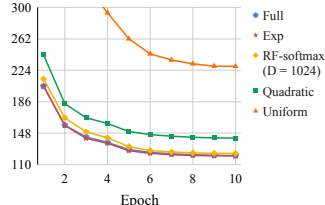

Figure 3: Comparison of RF-softmax with other baselines on PENNTREEBANK with $m = 100$ and validation perplexity as the metric.

**Effect of the parameter $\nu$.** As discussed in Section 3.2, for a finite $D$, we should choose $\nu < \tau$ as a trade off between the variance and the bias. Figure 1 shows the performance of the proposed

| Dataset | Method | PREC@1 | PREC@3 | PREC@5 |
|---------|--------|--------|--------|--------|
| AMAZONCAT-13K $n = 13,330$ $v = 203,882$ | EXP | 0.87 | 0.76 | 0.62 |
| | UNIFORM | 0.83 | 0.69 | 0.55 |
| | QUADRATIC | 0.84 | 0.74 | 0.60 |
| | RFF | 0.87 | 0.75 | 0.61 |
| DELICIOUS-200K $n = 205,443$ $v = 782,585$ | EXP | 0.42 | 0.38 | 0.37 |
| | UNIFORM | 0.36 | 0.34 | 0.32 |
| | QUADRATIC | 0.40 | 0.36 | 0.34 |
| | RFF | 0.41 | 0.37 | 0.36 |
| WIKILSHTC $n = 325,056$ $v = 1,617,899$ | EXP | 0.58 | 0.37 | 0.29 |
| | UNIFORM | 0.47 | 0.29 | 0.22 |
| | QUADRATIC | 0.57 | 0.37 | 0.28 |
| | RFF | 0.56 | 0.35 | 0.26 |

Table 3: Comparison among sampled softmax methods on extreme classification datasets. We report the metrics based on the same number of training iterations for all methods.

RF-softmax method on PENNTREEBANK for different values of $\nu$. In particular, we vary $T = \frac{1}{\sqrt{\nu}}$ as it defines the underlying RFF map (cf. (18)). The best performance is attained at $T = 0.5$. This choice of $\nu < \tau$ is in line with our discussion in Section 3.3. We use this same setting in the remaining experiments.

**Effect of $D$.** The accuracy of the approximation of the Gaussian kernel using the RFF map improves as we increase the dimension of the map $D$. Figure 2 demonstrates the performance of the proposed RF-softmax method on PENNTREEBANK for different values of $D$. As expected, the performance of RF-softmax gets closer to that of FULL when we increase $D$.

**RF-softmax vs. baselines.** Figure 3 illustrates the performance of different sampled softmax approaches on PEN-NTREEBANK. The figure shows the validation perplexity as the training progresses. As expected, the performance of expensive EXP is very close to the performance of FULL. RF-softmax outperforms both QUADRATIC and UNIFORM. We note that RF-softmax with $D = 1024$ performs better than QUADRATIC method at significantly lower computational and space cost. Since we have embedding dimension $d = 200$, RF-softmax is almost 40X more efficient as compared to QUADRATIC ($D$ vs. $d^2$). Figure 4 shows the performance of different methods on BNEWS. Note that the performance of RF-softmax is at par with QUADRATIC when $D = 2048$. Furthermore, RF-softmax outperforms QUADRATIC when $D = 8192$. In this experiment, we have $d = 512$, so RF-softmax with $D = 2048$ and $D = 8192$ are 128X and 32X more efficient than QUADRATIC, respectively.

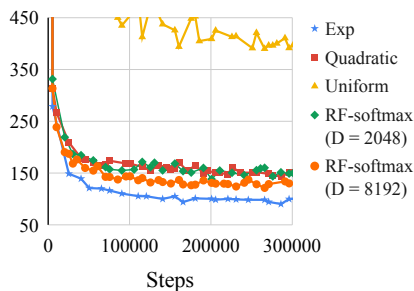

Figure 4: Comparison of RF-softmax with other baselines on BNEWS with $m = 100$ and validation perplexity as the metric.

Table 3 shows the performance of various sampling methods on three extreme classification datasets [33]. We do not report PREC@K values for FULL as the performance of EXP is an accurate proxy for those. The results demonstrate that RF-softmax attains better/comparable performance relative to QUADRATIC. For WIKILSHTC, even though QUADRATIC has better PREC@K values, RF-softmax leads to 4% smaller full softmax loss, which validates our analysis.

## Footnotes

[1]The results of this paper generalize to a multi-label setting by using multi-label to multi-class reductions [4].

[2]Since it is clear from the context, in what follows, we denote $\mathcal{L}(\mathbf{x}, t)$ and $\mathcal{L}'(\mathbf{x}, t)$ by $\mathcal{L}$ and $\mathcal{L}'$, respectively.

[3]This assumption naturally holds in most of the practical implementations, where each of the gradient coordinates or norm of the gradient is clipped by a threshold.

[4]This provides a wide enough range for the logits values; thus, ensuring the underlying model has sufficient expressive power. The typical choice ranges from 5 to 30.

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
