[Supplementary Material]

# A   Proof of Theorem 1

Before presenting the proof, we would like to point out that, unless specified otherwise, all the expectations in this section are taken over the sampling distribution $q$.

*Proof.* It follows from (8) that

$$\mathbb{E}\left[\nabla_{\boldsymbol{\theta}}\mathcal{L}'\right] = -\nabla_{\boldsymbol{\theta}}o_t + \mathbb{E}\left[\frac{e^{o_t}\cdot\nabla_{\boldsymbol{\theta}}o_t + \sum_{i\in[m]}\left(\frac{e^{o_{s_i}}}{mq_{s_i}}\cdot\nabla_{\boldsymbol{\theta}}o_{s_i}\right)}{e^{o_t} + \sum_{i\in[m]}\frac{e^{o_{s_i}}}{mq_{s_i}}}\right] \tag{21}$$

Let's define random variables

$$U = e^{o_t}\cdot\nabla_{\boldsymbol{\theta}}o_t + \sum_{i\in[m]}\left(\frac{e^{o_{s_i}}}{mq_{s_i}}\cdot\nabla_{\boldsymbol{\theta}}o_{s_i}\right) \tag{22}$$

and

$$V = e^{o_t} + \sum_{i\in[m]}\frac{e^{o_{s_i}}}{mq_{s_i}}. \tag{23}$$

Note that we have

$$\mathbb{E}[U] = \sum_{j\in[n]}e^{o_j}\cdot\nabla_{\boldsymbol{\theta}}o_j \ \text{ and } \ \mathbb{E}[V] = e^{o_t} + \sum_{j\in\mathcal{N}_t}e^{o_j} = \sum_{j\in[n]}e^{o_j} = Z, \tag{24}$$

**Lower bound:** It follows from (21) and the lower bound in Lemma 1 that

$$\mathbb{E}\left[\nabla_{\boldsymbol{\theta}}\mathcal{L}'\right] \geq -\nabla_{\boldsymbol{\theta}}o_t + \frac{e^{o_t}\cdot\nabla_{\boldsymbol{\theta}}o_t}{\sum_{j\in[n]}e^{o_j}} + \sum_{k\in\mathcal{N}_t}\frac{e^{o_k}\cdot\nabla_{\boldsymbol{\theta}}o_k}{e^{o_t} + \frac{m-1}{m}\cdot\sum_{j\in\mathcal{N}_t}e^{o_j} + \frac{e^{o_k}}{mq_k}}$$

$$= -\nabla_{\boldsymbol{\theta}}o_t + \frac{e^{o_t}\cdot\nabla_{\boldsymbol{\theta}}o_t}{\sum_{j\in[n]}e^{o_j}} +$$

$$\sum_{k\in\mathcal{N}_t}\frac{e^{o_k}\cdot\nabla_{\boldsymbol{\theta}}o_k}{\sum_{j\in[n]}e^{o_j}} + \left(\sum_{k\in\mathcal{N}_t}\frac{e^{o_k}\cdot\nabla_{\boldsymbol{\theta}}o_k}{e^{o_t} + \frac{m-1}{m}\cdot\sum_{j\in\mathcal{N}_t}e^{o_j} + \frac{e^{o_k}}{mq_k}} - \sum_{k\in\mathcal{N}_t}\frac{e^{o_k}\cdot\nabla_{\boldsymbol{\theta}}o_k}{e^{o_t} + \sum_{j\in\mathcal{N}_t}e^{o_j}}\right)$$

$$= -\nabla_{\boldsymbol{\theta}}o_t + \frac{e^{o_t}\cdot\nabla_{\boldsymbol{\theta}}o_t}{\sum_{j\in[n]}e^{o_j}} + \frac{\sum_{k\in\mathcal{N}_t}e^{o_k}\cdot\nabla_{\boldsymbol{\theta}}o_k}{\sum_{j\in[n]}e^{o_j}} +$$

$$\frac{1}{m}\sum_{k\in\mathcal{N}_t}\frac{e^{o_k}\cdot\left(\sum_{j\in\mathcal{N}_t}e^{o_j} - \frac{e^{o_k}}{q_k}\right)\cdot\nabla_{\boldsymbol{\theta}}o_k}{\left(e^{o_t} + \frac{m-1}{m}\cdot\sum_{j\in\mathcal{N}_t}e^{o_j} + \frac{e^{o_k}}{mq_k}\right)\left(e^{o_t} + \sum_{j\in\mathcal{N}_t}e^{o_j}\right)}$$

$$= \nabla_{\boldsymbol{\theta}}\mathcal{L} + \frac{1}{m}\sum_{k\in\mathcal{N}_t}\frac{e^{o_k}\cdot\left(\sum_{j\in\mathcal{N}_t}e^{o_j} - \frac{e^{o_k}}{q_k}\right)\cdot\nabla_{\boldsymbol{\theta}}o_k}{\left(e^{o_t} + \frac{m-1}{m}\cdot\sum_{j\in\mathcal{N}_t}e^{o_j} + \frac{e^{o_k}}{mq_k}\right)\left(e^{o_t} + \sum_{j\in\mathcal{N}_t}e^{o_j}\right)}$$

$$\geq \nabla_{\boldsymbol{\theta}}\mathcal{L} - \frac{1}{m}\sum_{k\in\mathcal{N}_t}\frac{e^{o_k}\cdot\left|\sum_{j\in\mathcal{N}_t}e^{o_j} - \frac{e^{o_k}}{q_k}\right|\cdot\left|\nabla_{\boldsymbol{\theta}}o_k\right|}{\left(e^{o_t} + \frac{m-1}{m}\cdot\sum_{j\in\mathcal{N}_t}e^{o_j} + \frac{e^{o_k}}{mq_k}\right)\left(e^{o_t} + \sum_{j\in\mathcal{N}_t}e^{o_j}\right)}$$

$$\overset{(i)}{\geq} \nabla_{\boldsymbol{\theta}}\mathcal{L} - \frac{M}{m}\sum_{k\in\mathcal{N}_t}\frac{e^{o_k}\cdot\left|\sum_{j\in\mathcal{N}_t}e^{o_j} - \frac{e^{o_k}}{q_k}\right|}{\left(e^{o_t} + \frac{m-1}{m}\cdot\sum_{j\in\mathcal{N}_t}e^{o_j}\right)\left(e^{o_t} + \sum_{j\in\mathcal{N}_t}e^{o_j}\right)}\cdot\mathbf{1}$$

$$= \nabla_{\boldsymbol{\theta}}\mathcal{L} - \frac{M}{m}\sum_{k\in\mathcal{N}_t}\frac{e^{o_k}\cdot\left|\sum_{j\in\mathcal{N}_t}e^{o_j} - \frac{e^{o_k}}{q_k}\right|}{\left(e^{o_t} + \sum_{j\in\mathcal{N}_t}e^{o_j}\right)^2}\left(1 - o\left(\frac{1}{m}\right)\right)\cdot\mathbf{1}$$

$$= \nabla_{\boldsymbol{\theta}}\mathcal{L} - \frac{M}{m}\sum_{k\in\mathcal{N}_t}\frac{e^{o_k}\cdot\left|\sum_{j\in\mathcal{N}_t}e^{o_j} - \frac{e^{o_k}}{q_k}\right|}{Z^2}\left(1 - o\left(\frac{1}{m}\right)\right)\cdot\mathbf{1}, \tag{25}$$

where $(i)$ follows from the bound on the entries of $\nabla_{\boldsymbol{\theta}} o_k$, for each $k \in [n]$.

**Upper bound:** Using (21) and the upper bound in Lemma 1, we obtain that

$$\mathbb{E}\left[\nabla_{\boldsymbol{\theta}} \mathcal{L}'\right] \leq -\nabla_{\boldsymbol{\theta}} o_t + \mathbb{E}[U] \cdot \mathbb{E}\left[\frac{1}{V}\right] + \Delta_m, \tag{26}$$

where

$$\Delta_m \triangleq \frac{1}{m}\mathbb{E}\left[\sum_{k \in \mathcal{N}_t} \frac{e^{o_k}\left|\nabla_{\boldsymbol{\theta}} o_k\right| \cdot \left|\frac{e^{o_{s_m}}}{q_{s_m}} - \frac{e^{o_k}}{q_k}\right|}{\left(e^{o_t} + \sum_{i \in [m-1]} \frac{e^{o_{s_i}}}{m q_{s_i}}\right)^2}\right].$$

By employing Lemma 4 with (26), we obtain the following.

$$\mathbb{E}\left[\nabla_{\boldsymbol{\theta}} \mathcal{L}'\right] \leq -\nabla_{\boldsymbol{\theta}} o_t + \frac{\mathbb{E}[U]}{\mathbb{E}[V]} + \mathbb{E}[U] \cdot \left(\frac{\sum_{j \in \mathcal{N}_t} \frac{e^{2o_j}}{q_j} - \left(\sum_{j \in \mathcal{N}_t} e^{o_j}\right)^2}{m Z^3} + o\left(\frac{1}{m}\right)\right) + \Delta_m$$

$$= \nabla_{\boldsymbol{\theta}} \mathcal{L} + \mathbb{E}[U] \cdot \left(\frac{\sum_{j \in \mathcal{N}_t} \frac{e^{2o_j}}{q_j} - \left(\sum_{j \in \mathcal{N}_t} e^{o_j}\right)^2}{m Z^3} + o\left(\frac{1}{m}\right)\right) + \Delta_m \tag{27}$$

By employing Lemma 2 in (27), we obtain that

$$\mathbb{E}\left[\nabla_{\boldsymbol{\theta}} \mathcal{L}'\right] \leq \nabla_{\boldsymbol{\theta}} \mathcal{L} + \mathbb{E}[U] \cdot \left(\frac{\sum_{j \in \mathcal{N}_t} \frac{e^{2o_j}}{q_j} - \left(\sum_{j \in \mathcal{N}_t} e^{o_j}\right)^2}{m Z^3} + o\left(\frac{1}{m}\right)\right) + $$

$$\left(\frac{2M}{m} \frac{\max_{i,i' \in \mathcal{N}_t}\left|\frac{e^{o_i}}{q_i} - \frac{e^{o_{i'}}}{q_{i'}}\right| \sum_{j \in \mathcal{N}_t} e^{o_j}}{Z^2 + \sum_{j \in \mathcal{N}_t} \frac{e^{2o_j}}{q_j}} + o\left(\frac{1}{m}\right)\right) \cdot \mathbf{1}$$

Now, using $\mathbb{E}[U] = \sum_{j \in [n]} e^{o_j} \cdot \nabla_{\boldsymbol{\theta}} o_j$ gives the stated upper bound. $\qquad\square$

**Lemma 1.** *Let $\mathcal{S} = \{s_1, \ldots, s_m\} \subset \mathcal{N}_t^m$ be $m$ i.i.d. negative classes drawn according to the sampling distribution $q$. Then, the ratio appearing in the gradient estimate based on the sample softmax approach (cf.* (8)) *satisfies*

$$\frac{e^{o_t} \cdot \nabla_{\boldsymbol{\theta}} o_t}{\sum_{j \in [n]} e^{o_j}} + \sum_{k \in \mathcal{N}_t} \frac{e^{o_k} \cdot \nabla_{\boldsymbol{\theta}} o_k}{e^{o_t} + \frac{m-1}{m} \cdot \sum_{j \in \mathcal{N}_t} e^{o_j} + \frac{e^{o_k}}{m q_k}} \leq$$

$$\mathbb{E}\left[\frac{e^{o_t} \cdot \nabla_{\boldsymbol{\theta}} o_t + \sum_{i \in [m]}\left(\frac{e^{o_{s_i}}}{m q_{s_i}} \cdot \nabla_{\boldsymbol{\theta}} o_s\right)}{e^{o_t} + \sum_{i \in [m]} \frac{e^{o_{s_i}}}{m q_{s_i}}}\right] \leq$$

$$\left(\sum_{k \in [n]} e^{o_k} \cdot \nabla_{\boldsymbol{\theta}} o_k\right) \cdot \mathbb{E}\left[\frac{1}{e^{o_t} + \sum_{i \in [m]} \frac{e^{o_{s_i}}}{m q_{s_i}}}\right] + \Delta_m, \tag{28}$$

*where*

$$\Delta_m \triangleq \frac{1}{m}\mathbb{E}\left[\sum_{k \in \mathcal{N}_t} \frac{e^{o_k}\left|\nabla_{\boldsymbol{\theta}} o_k\right| \cdot \left|\frac{e^{o_{s_m}}}{q_{s_m}} - \frac{e^{o_k}}{q_k}\right|}{\left(e^{o_t} + \sum_{i \in [m-1]} \frac{e^{o_{s_i}}}{m q_{s_i}}\right)^2}\right] \tag{29}$$

*Proof.* Note that

$$\mathbb{E}\left[\frac{e^{o_t} \cdot \nabla_{\boldsymbol{\theta}} o_t + \sum_{i \in [m]}\left(\frac{e^{o_{s_i}}}{m q_{s_i}} \cdot \nabla_{\boldsymbol{\theta}} o_s\right)}{e^{o_t} + \sum_{i \in [m]} \frac{e^{o_{s_i}}}{m q_{s_i}}}\right]$$

$$= \mathbb{E}\left[\frac{e^{o_t} \cdot \nabla_{\boldsymbol{\theta}} o_t}{e^{o_t} + \sum_{j \in [m]} \frac{e^{o_{s_j}}}{m q_{s_j}}}\right] + \sum_{i \in [m]} \mathbb{E}\left[\frac{\left(\frac{e^{o_{s_i}}}{m q_{s_i}} \cdot \nabla_{\boldsymbol{\theta}} o_{s_i}\right)}{e^{o_t} + \sum_{j \in [m]} \frac{e^{o_{s_j}}}{m q_{s_j}}}\right]$$

$$= \mathbb{E}\left[\frac{e^{o_t} \cdot \nabla_{\boldsymbol{\theta}} o_t}{e^{o_t} + \sum_{j\in[m]} \frac{e^{o_{s_j}}}{mq_{s_j}}}\right] + m \cdot \underbrace{\mathbb{E}\left[\frac{\left(\frac{e^{o_{s_m}}}{mq_{s_m}} \cdot \nabla_{\boldsymbol{\theta}} o_{s_m}\right)}{e^{o_t} + \sum_{j\in[m]} \frac{e^{o_{s_j}}}{mq_{s_j}}}\right]}_{\text{Term I}} \qquad (30)$$

For $1 \le l \le m$, let's define the notation $S_l \triangleq \sum_{j\in[l]} \frac{e^{o_{s_j}}}{q_{s_j}}$. Now, let's consider Term I.

$$\text{Term I} = \mathbb{E}\left[\mathbb{E}\left[\frac{\frac{e^{o_{s_m}}}{mq_{s_m}} \cdot \nabla_{\boldsymbol{\theta}} o_{s_m}}{e^{o_t} + \frac{S_{m-1}}{m} + \frac{e^{o_{s_m}}}{mq_{s_m}}} \,\Big|\, S_{m-1}\right]\right]$$

$$= \mathbb{E}\left[\sum_{k\in\mathcal{N}_t} q_k \cdot \frac{\frac{e^{o_k}}{mq_k} \cdot \nabla_{\boldsymbol{\theta}} o_k}{e^{o_t} + \frac{S_{m-1}}{m} + \frac{e^{o_k}}{mq_k}}\right]$$

$$= \frac{1}{m} \sum_{k\in\mathcal{N}_t} \mathbb{E}\left[\frac{e^{o_k} \cdot \nabla_{\boldsymbol{\theta}} o_k}{e^{o_t} + \frac{S_{m-1}}{m} + \frac{e^{o_k}}{mq_k}}\right] \qquad (31)$$

$$= \frac{1}{m} \sum_{k\in\mathcal{N}_t} \mathbb{E}\left[\frac{e^{o_k} \cdot \nabla_{\boldsymbol{\theta}} o_k}{e^{o_t} + \frac{S_{m-1}}{m} + \frac{e^{o_k}}{mq_k}} - \frac{e^{o_k} \cdot \nabla_{\boldsymbol{\theta}} o_k}{e^{o_t} + \frac{S_m}{m}} + \frac{e^{o_k} \cdot \nabla_{\boldsymbol{\theta}} o_k}{e^{o_t} + \frac{S_m}{m}}\right] \qquad (32)$$

$$= \frac{1}{m} \sum_{k\in\mathcal{N}_t} \mathbb{E}\left[\frac{e^{o_k} \cdot \nabla_{\boldsymbol{\theta}} o_k}{e^{o_t} + \frac{S_{m-1}}{m} + \frac{e^{o_k}}{mq_k}} - \frac{e^{o_k} \cdot \nabla_{\boldsymbol{\theta}} o_k}{e^{o_t} + \frac{S_m}{m}}\right] + \frac{1}{m} \sum_{k\in\mathcal{N}_t} e^{o_k} \cdot \nabla_{\boldsymbol{\theta}} o_k \cdot \mathbb{E}\left[\frac{1}{e^{o_t} + \frac{S_m}{m}}\right]$$

$$\le \frac{1}{m^2} \mathbb{E}\left[\sum_{k\in\mathcal{N}_t} \frac{e^{o_k}|\nabla_{\boldsymbol{\theta}} o_k| \cdot \left|\frac{e^{o_{s_m}}}{q_{s_m}} - \frac{e^{o_k}}{q_k}\right|}{\left(e^{o_t} + \frac{S_{m-1}}{m} + \frac{e^{o_k}}{mq_k}\right)\left(e^{o_t} + \frac{S_m}{m}\right)}\right] + \frac{1}{m} \sum_{k\in\mathcal{N}_t} e^{o_k} \cdot \nabla_{\boldsymbol{\theta}} o_k \cdot \mathbb{E}\left[\frac{1}{e^{o_t} + \frac{S_m}{m}}\right]$$

$$\overset{(i)}{\le} \frac{1}{m^2} \mathbb{E}\left[\sum_{k\in\mathcal{N}_t} \frac{e^{o_k}|\nabla_{\boldsymbol{\theta}} o_k| \cdot \left|\frac{e^{o_{s_m}}}{q_{s_m}} - \frac{e^{o_k}}{q_k}\right|}{\left(e^{o_t} + \frac{S_{m-1}}{m}\right)^2}\right] + \frac{1}{m} \sum_{k\in\mathcal{N}_t} e^{o_k} \cdot \nabla_{\boldsymbol{\theta}} o_k \cdot \mathbb{E}\left[\frac{1}{e^{o_t} + \frac{S_m}{m}}\right], \qquad (33)$$

where $(i)$ follows by dropping the positive terms $\frac{e^{o_k}}{mq_k}$ and $\frac{e^{o_{s_i}}}{mq_{s_i}}$. Next, we combine(30) and (33) to obtain the following.

$$\mathbb{E}\left[\frac{e^{o_t} \cdot \nabla_{\boldsymbol{\theta}} o_t + \sum_{i\in[m]}\left(\frac{e^{o_{s_i}}}{mq_{s_i}} \cdot \nabla_{\boldsymbol{\theta}} o_s\right)}{e^{o_t} + \sum_{i\in[m]} \frac{e^{o_{s_i}}}{mq_{s_i}}}\right]$$

$$\le e^{o_t} \cdot \nabla_{\boldsymbol{\theta}} o_t \cdot \mathbb{E}\left[\frac{1}{e^{o_t} + \frac{S_m}{m}}\right] + \sum_{k\in\mathcal{N}_t} e^{o_k} \cdot \nabla_{\boldsymbol{\theta}} o_k \cdot \mathbb{E}\left[\frac{1}{e^{o_t} + \frac{S_m}{m}}\right] +$$

$$\frac{1}{m}\mathbb{E}\left[\sum_{k\in\mathcal{N}_t} \frac{e^{o_k}|\nabla_{\boldsymbol{\theta}} o_k| \cdot \left|\frac{e^{o_{s_m}}}{q_{s_m}} - \frac{e^{o_k}}{q_k}\right|}{\left(e^{o_t} + \frac{S_{m-1}}{m}\right)^2}\right]$$

$$= \left(\sum_{k\in[n]} e^{o_k} \cdot \nabla_{\boldsymbol{\theta}} o_k\right) \cdot \mathbb{E}\left[\frac{1}{e^{o_t} + \sum_{i\in[m]} \frac{e^{o_{s_i}}}{mq_{s_i}}}\right] + \frac{1}{m}\mathbb{E}\left[\sum_{k\in\mathcal{N}_t} \frac{e^{o_k}|\nabla_{\boldsymbol{\theta}} o_k| \cdot \left|\frac{e^{o_{s_m}}}{q_{s_m}} - \frac{e^{o_k}}{q_k}\right|}{\left(e^{o_t} + \frac{S_{m-1}}{m}\right)^2}\right]$$

$$= \left(\sum_{k\in[n]} e^{o_k} \cdot \nabla_{\boldsymbol{\theta}} o_k\right) \cdot \mathbb{E}\left[\frac{1}{e^{o_t} + \sum_{i\in[m]} \frac{e^{o_{s_i}}}{mq_{s_i}}}\right] + \Delta_m. \qquad (34)$$

This completes the proof of the upper bound in (28). In order to establish the lower bound in (28), we combine (30) and (31) to obtain that

$$\mathbb{E}\left[\frac{e^{o_t} \cdot \nabla_{\boldsymbol{\theta}} o_t + \sum_{i\in[m]}\left(\frac{e^{o_{s_i}}}{mq_{s_i}} \cdot \nabla_{\boldsymbol{\theta}} o_s\right)}{e^{o_t} + \sum_{i\in[m]} \frac{e^{o_{s_i}}}{mq_{s_i}}}\right]$$

$$= \mathbb{E}\left[\frac{e^{o_t} \cdot \nabla_{\boldsymbol{\theta}} o_t}{e^{o_t} + \sum_{j \in [m]} \frac{e^{o_{s_j}}}{m q_{s_j}}}\right] + \sum_{k \in \mathcal{N}_t} \mathbb{E}\left[\frac{e^{o_k} \cdot \nabla_{\boldsymbol{\theta}} o_k}{e^{o_t} + \frac{S_{m-1}}{m} + \frac{e^{o_k}}{m q_k}}\right]$$

$$\geq \frac{e^{o_t} \cdot \nabla_{\boldsymbol{\theta}} o_t}{\sum_{j \in [n]} e^{o_j}} + \sum_{k \in \mathcal{N}_t} \frac{e^{o_k} \cdot \nabla_{\boldsymbol{\theta}} o_k}{e^{o_t} + \frac{m-1}{m} \cdot \sum_{j \in \mathcal{N}_t} e^{o_j} + \frac{e^{o_k}}{m q_k}}, \tag{35}$$

where the final step follows by applying Jensen's inequality to each of the expectation terms. $\quad\square$

**Lemma 2.** *For any model parameter $\boldsymbol{\theta} \in \Theta$, assume that we have the following bound on the maximum absolute value of each of coordinates of the gradient vectors*

$$\|\nabla_{\boldsymbol{\theta}} o_j\|_\infty \leq M \quad \forall j \in [n]. \tag{36}$$

*Then, $\Delta_m$ defined in Lemma 1 satisfies*

$$\Delta_m \triangleq \frac{1}{m} \mathbb{E}\left[\sum_{k \in \mathcal{N}_t} \frac{e^{o_k} |\nabla_{\boldsymbol{\theta}} o_k| \cdot \left|\frac{e^{o_{s_m}}}{q_{s_m}} - \frac{e^{o_k}}{q_k}\right|}{\left(e^{o_t} + \sum_{i \in [m-1]} \frac{e^{o_{s_i}}}{m q_{s_i}}\right)^2}\right]$$

$$\leq \left(\frac{2M}{m} \frac{\max_{i,i' \in \mathcal{N}_t} \left|\frac{e^{o_i}}{q_i} - \frac{e^{o_{i'}}}{q_{i'}}\right| \sum_{j \in \mathcal{N}_t} e^{o_j}}{Z^2 + \sum_{j \in \mathcal{N}_t} \frac{e^{2o_j}}{q_j}} + o\left(\frac{1}{m}\right)\right) \cdot \mathbf{1}, \tag{37}$$

*where $\mathbf{1}$ is the all one vector.*

*Proof.* Note that

$$\Delta_m \triangleq \frac{1}{m} \mathbb{E}\left[\sum_{k \in \mathcal{N}_t} \frac{e^{o_k} |\nabla_{\boldsymbol{\theta}} o_k| \cdot \left|\frac{e^{o_{s_m}}}{q_{s_m}} - \frac{e^{o_k}}{q_k}\right|}{\left(e^{o_t} + \sum_{i \in [m-1]} \frac{e^{o_{s_i}}}{m q_{s_i}}\right)^2}\right]$$

$$\leq \frac{1}{m} \mathbb{E}\left[\sum_{k \in \mathcal{N}_t} \frac{e^{o_k} |\nabla_{\boldsymbol{\theta}} o_k| \cdot \max_{i,i' \in \mathcal{N}_t} \left|\frac{e^{o_i}}{q_i} - \frac{e^{o_{i'}}}{q_{i'}}\right|}{\left(e^{o_t} + \sum_{i \in [m-1]} \frac{e^{o_{s_i}}}{m q_{s_i}}\right)^2}\right]$$

$$\leq \frac{2 \cdot \max_{i,i' \in \mathcal{N}_t} \left|\frac{e^{o_i}}{q_i} - \frac{e^{o_{i'}}}{q_{i'}}\right|}{m} \cdot \mathbb{E}\left[\frac{1}{\left(e^{o_t} + \frac{S_{m-1}}{m}\right)^2}\right] \cdot \sum_{k \in \mathcal{N}_t} e^{o_k} |\nabla_{\boldsymbol{\theta}} o_k|$$

$$\overset{(i)}{\leq} \frac{2M \cdot \max_{i,i' \in \mathcal{N}_t} \left|\frac{e^{o_i}}{q_i} - \frac{e^{o_{i'}}}{q_{i'}}\right|}{m} \cdot \mathbb{E}\left[\frac{1}{\left(e^{o_t} + \frac{S_{m-1}}{m}\right)^2}\right] \cdot \left(\sum_{k \in \mathcal{N}_t} e^{o_k}\right) \cdot \mathbf{1}, \tag{38}$$

where $(i)$ follows from the fact that the gradients have bounded entries. Now, combining (38) and Lemma 3 gives us that

$$\Delta_m \leq \left(\frac{2M}{m} \frac{\max_{i,i' \in \mathcal{N}_t} \left|\frac{e^{o_i}}{q_i} - \frac{e^{o_{i'}}}{q_{i'}}\right| \sum_{j \in \mathcal{N}_t} e^{o_j}}{Z^2 + \sum_{j \in \mathcal{N}_t} \frac{e^{2o_j}}{q_j}} + o\left(\frac{1}{m}\right)\right) \cdot \mathbf{1}. \tag{39}$$

$\square$

**Lemma 3.** *Consider the random variable*

$$W = \left(e^{o_t} + \frac{S_{m-1}}{m}\right)^2. \tag{40}$$

*Then, we have*

$$\mathbb{E}\left[\frac{1}{W}\right] = \frac{1}{Z^2 + \sum_{j \in \mathcal{N}_t} \frac{e^{2o_j}}{q_j}} + \mathcal{O}\left(\frac{1}{m}\right). \tag{41}$$

*Proof.* Note that

$$\mathbb{E}[W] = \mathbb{E}\left[\left(e^{o_t} + \frac{S_{m-1}}{m}\right)^2\right]$$

$$= e^{2o_t} + \frac{2e^{o_t}}{m}\sum_{i\in[m-1]}\mathbb{E}\left[\frac{e^{o_{s_i}}}{q_{s_i}}\right] + \frac{1}{m^2}\mathbb{E}\left[\sum_{i,i'\in[m-1]}\frac{e^{o_{s_i}+o_{s_{i'}}}}{q_{s_i}q_{s_{i'}}}\right]$$

$$= e^{2o_t} + \frac{m-1}{m}\cdot 2e^{o_t}\sum_{j\in\mathcal{N}_t}e^{o_j} + \frac{1}{m^2}\mathbb{E}\left[\sum_{i,i'\in[m-1]}\frac{e^{o_{s_i}+o_{s_{i'}}}}{q_{s_i}q_{s_{i'}}}\right]$$

$$= e^{2o_t} + \frac{m-1}{m}\cdot 2e^{o_t}\sum_{j\in\mathcal{N}_t}e^{o_j} + \frac{m-1}{m}\cdot\sum_{j\in\mathcal{N}_t}\frac{e^{2o_j}}{q_j} + \frac{(m-1)(m-2)}{m^2}\cdot\sum_{j,j'\in\mathcal{N}_t}e^{o_j+o_{j'}}$$

$$= Z^2 + \sum_{j\in\mathcal{N}_t}\frac{e^{2o_j}}{q_j} - \frac{1}{m}\cdot 2e^{o_t}\sum_{j\in\mathcal{N}_t}e^{o_j} - \frac{1}{m}\cdot\sum_{j\in\mathcal{N}_t}\frac{e^{2o_j}}{q_j} - \frac{3m-2}{m^2}\cdot\sum_{j,j'\in\mathcal{N}_t}e^{o_j+o_{j'}}$$

$$\triangleq \overline{W}_m. \tag{42}$$

Furthermore, one can verify that $\mathrm{Var}(W)$ scale as $\mathcal{O}\left(\frac{1}{m}\right)$. Therefore, using (59) in Lemma 5, we have

$$\mathbb{E}\left[\frac{1}{W}\right] \le \frac{1}{\mathbb{E}[W]} + \frac{\mathrm{Var}(W)}{e^{6o_t}} = \frac{1}{\mathbb{E}[W]} + \mathcal{O}\left(\frac{1}{m}\right). \tag{43}$$

Now, it follows from (42) and (43) that

$$\mathbb{E}\left[\frac{1}{W}\right] \le \frac{1}{\overline{W}_m} + \mathcal{O}\left(\frac{1}{m}\right),$$

which with some additional algebra can be shown to be

$$\mathbb{E}\left[\frac{1}{W}\right] = \frac{1}{Z^2 + \sum_{j\in\mathcal{N}_t}\frac{e^{2o_j}}{q_j}} + \mathcal{O}\left(\frac{1}{m}\right).$$

$\square$

**Lemma 4.** *Consider the random variable*

$$V = e^{o_t} + \sum_{i\in[m]}\frac{e^{o_{s_i}}}{mq_{s_i}}. \tag{44}$$

*Then, we have*

$$\mathbb{E}\left[\frac{1}{V}\right] \le \frac{1}{\mathbb{E}[V]} + \frac{\sum_{j\in\mathcal{N}_t}\frac{e^{2o_j}}{q_j} - \left(\sum_{j\in\mathcal{N}_t}e^{o_j}\right)^2}{mZ^3} + o\left(\frac{1}{m}\right). \tag{45}$$

*Proof.* We have from Lemma 5 that

$$\mathbb{E}\left[\frac{1}{V}\right] \le \frac{1}{\mathbb{E}[V]} + \frac{\mathrm{Var}(V)}{\mathbb{E}[V]^3} + o\left(\frac{1}{m}\right). \tag{46}$$

Note that

$$\mathbb{E}[V^2] = e^{2o_t} + \frac{e^{o_t}}{m}\mathbb{E}\left[\sum_{i\in[m]}\frac{2e^{o_{s_i}}}{q_{s_i}}\right] + \frac{1}{m^2}\cdot\mathbb{E}\left[\sum_{i,i'\in[m]}\frac{e^{o_{s_i}+o_{s_{i'}}}}{q_{s_i}q_{s_{i'}}}\right]$$

$$= e^{2o_t} + 2e^{o_t}\cdot\sum_{j\in\mathcal{N}_t}e^{o_j} + \frac{1}{m^2}\cdot\mathbb{E}\left[\sum_{i\in[m]}\frac{e^{2o_{s_i}}}{q_{s_i}^2} + \sum_{i\ne i'\in[m]}\frac{e^{o_{s_i}+o_{s_{i'}}}}{q_{s_i}q_{s_{i'}}}\right]$$

$$= e^{2o_t} + 2e^{o_t} \cdot \sum_{j \in \mathcal{N}_t} e^{o_j} + \frac{1}{m} \cdot \sum_{j \in \mathcal{N}_t} \frac{e^{2o_j}}{q_j} + \frac{m-1}{m} \cdot \sum_{j,j' \in \mathcal{N}_t} e^{o_j + o_{j'}}$$

$$= e^{2o_t} + 2e^{o_t} \cdot \sum_{j \in \mathcal{N}_t} e^{o_j} + \sum_{j,j' \in \mathcal{N}_t} e^{o_j + o_{j'}} + \frac{1}{m} \cdot \sum_{j \in \mathcal{N}_t} \frac{e^{2o_j}}{q_j} - \frac{1}{m} \cdot \sum_{j,j' \in \mathcal{N}_t} e^{o_j + o_{j'}}$$

$$= \left( \sum_{j \in [n]} e^{o_j} \right)^2 + \frac{1}{m} \cdot \sum_{j \in \mathcal{N}_t} \frac{e^{2o_j}}{q_j} - \frac{1}{m} \cdot \sum_{j,j' \in \mathcal{N}_t} e^{o_j + o_{j'}}$$

$$= \mathbb{E}[V]^2 + \frac{1}{m} \cdot \sum_{j \in \mathcal{N}_t} \frac{e^{2o_j}}{q_j} - \frac{1}{m} \cdot \sum_{j,j' \in \mathcal{N}_t} e^{o_j + o_{j'}}$$

$$(47)$$

Therefore, we have

$$\mathrm{Var}(V) = \mathbb{E}[V^2] - \mathbb{E}[V]^2 = \frac{1}{m} \cdot \sum_{j \in \mathcal{N}_t} \frac{e^{2o_j}}{q_j} - \frac{1}{m} \cdot \sum_{j,j' \in \mathcal{N}_t} e^{o_j + o_{j'}} \tag{48}$$

Now, by combining (46) and (48) we obtain that

$$\mathbb{E}\left[ \frac{1}{V} \right] \leq \frac{1}{\mathbb{E}[V]} + \frac{\sum_{j \in \mathcal{N}_t} \frac{e^{2o_j}}{q_j} - \sum_{j,j' \in \mathcal{N}_t} e^{o_j + o_{j'}}}{mZ^3} + o\left( \frac{1}{m} \right)$$

$$= \frac{1}{\mathbb{E}[V]} + \frac{\sum_{j \in \mathcal{N}_t} \frac{e^{2o_j}}{q_j} - \left( \sum_{j \in \mathcal{N}_t} e^{o_j} \right)^2}{mZ^3} + o\left( \frac{1}{m} \right).$$

$\square$

## B  Proof of Theorem 2

*Proof.* Recall that RFFs provide an unbiased estimate of the Gaussian kernel. Therefore, we have

$$\mathbb{E}\left[ \phi(\mathbf{c}_i)^T \phi(\mathbf{h}) \right] = e^{-\nu \|\mathbf{c}_i - \mathbf{h}\|^2 / 2} = e^{-\nu} \cdot e^{\nu \mathbf{c}_i^T \mathbf{h}}. \tag{49}$$

In fact, for large enough $D$, the RFFs provide a tight approximation of $e^{-\nu} \cdot \exp(\nu \mathbf{c}_i^T \mathbf{h})$ [21, 25]. This follows from the observation that

$$\phi(\mathbf{c}_i)^T \phi(\mathbf{h}) = \frac{1}{\sqrt{D}} \sum_{j=1}^{D} \cos \left( \mathbf{w}_j^T (\mathbf{c}_i - \mathbf{h}) \right) \tag{50}$$

is a sum of $D$ bounded random variables

$$\left\{ \cos \left( \mathbf{w}_j^T (\mathbf{c}_i - \mathbf{h}) \right) \right\}_{j \in [D]}.$$

Here, $\mathbf{w}_1, \ldots, \mathbf{w}_D$ are i.i.d. random variable distributed according to the normal distribution $N(0, \nu \mathbf{I})$. Therefore, the following holds for all $\ell_2$-normalized vectors $\mathbf{u}, \mathbf{v} \in \mathbb{R}^d$ with probability at least $1 - \mathcal{O}\left( \frac{1}{D^2} \right)$ [21, 25].

$$\left| \phi(\mathbf{u})^T \phi(\mathbf{v}) - e^{-\nu} \cdot e^{\nu \mathbf{u}^T \mathbf{v}} \right| \leq \rho \sqrt{\frac{d}{D} \log(D)}, \tag{51}$$

where $\rho > 0$ is a constant. Note that we have

$$q_i = \frac{\phi(\mathbf{c}_i)^T \phi(\mathbf{h})}{\sum_{j \in \mathcal{N}_t} \phi(\mathbf{c}_j)^T \phi(\mathbf{h})} = \frac{1}{C} \cdot \phi(\mathbf{c}_i)^T \phi(\mathbf{h}) \ \forall i \in \mathcal{N}_t, \tag{52}$$

where the input embedding $\mathbf{h}$ and the class embeddings $\mathbf{c}_1, \ldots, \mathbf{c}_n$ are $\ell_2$-normalized. Therefore, it follows from (51) that the following holds with probability at least $1 - \mathcal{O}\left( \frac{1}{D^2} \right)$.

$$\frac{e^{\tau \mathbf{c}_i^T \mathbf{h}}}{e^{-\nu} e^{\nu \mathbf{c}_i^T \mathbf{h}} + \rho \sqrt{\frac{d}{D} \log(D)}} \leq \frac{1}{C} \cdot \left| \frac{e^{o_i}}{q_i} \right| \leq \frac{e^{\tau \mathbf{c}_i^T \mathbf{h}}}{e^{-\nu} e^{\nu \mathbf{c}_i^T \mathbf{h}} - \rho \sqrt{\frac{d}{D} \log(D)}}$$

or

$$\frac{e^{(\tau-\nu)\mathbf{c}_i^T\mathbf{h}}}{1 + \rho\sqrt{\frac{d}{D}}\log(D) \cdot e^{\nu(1-\mathbf{c}_i^T\mathbf{h})}} \leq \frac{1}{Ce^\nu} \cdot \left|\frac{e^{o_i}}{q_i}\right| \leq \frac{e^{(\tau-\nu)\mathbf{c}_i^T\mathbf{h}}}{1 - \rho\sqrt{\frac{d}{D}}\log(D) \cdot e^{\nu(1-\mathbf{c}_i^T\mathbf{h})}}. \tag{53}$$

Since we assume that both the input and the class embedding are normalized, we have

$$\exp(\nu\mathbf{c}_i^T\mathbf{h}) \in [e^{-\nu}, e^\nu]. \tag{54}$$

Thus, as long as, we ensure that

$$\rho\sqrt{\frac{d}{D}}\log(D) \cdot e^{2\nu} \leq \gamma \ \text{ or } \ e^{2\nu} \leq \frac{\gamma}{\rho\sqrt{d}} \cdot \frac{\sqrt{D}}{\log D} \tag{55}$$

for a small enough constant $\gamma > 0$, it follows from (53) that

$$\frac{e^{(\tau-\nu)\mathbf{c}_i^T\mathbf{h}}}{1 + \gamma} \leq \frac{1}{Ce^\nu} \cdot \left|\frac{e^{o_i}}{q_i}\right| \leq \frac{e^{(\tau-\nu)\mathbf{c}_i^T\mathbf{h}}}{1 - \gamma}. \tag{56}$$

Using the similar arguments, it also follows from (51) that with probability at least $1 - \mathcal{O}\left(\frac{1}{D^2}\right)$, we have

$$(1 - \gamma) \cdot e^{-\nu} \cdot \sum_{i\in\mathcal{N}_t} e^{o_i} \leq C \leq (1 + \gamma) \cdot e^{-\nu} \cdot \sum_{i\in\mathcal{N}_t} e^{o_i} \tag{57}$$

Now, by combining (56) and (57), we get that

$$e^{(\tau-\nu)\mathbf{c}_i^T\mathbf{h}} \cdot \frac{1 - \gamma}{1 + \gamma} \leq \frac{1}{\sum_{i\in\mathcal{N}_t} e^{o_i}} \cdot \left|\frac{e^{o_i}}{q_i}\right| \leq e^{(\tau-\nu)\mathbf{c}_i^T\mathbf{h}} \cdot \frac{1 + \gamma}{1 - \gamma}$$

or

$$e^{(\tau-\nu)\mathbf{c}_i^T\mathbf{h}} \cdot (1 - 2\gamma) \leq \frac{1}{\sum_{i\in\mathcal{N}_t} e^{o_i}} \cdot \left|\frac{e^{o_i}}{q_i}\right| \leq e^{(\tau-\nu)\mathbf{c}_i^T\mathbf{h}} \cdot (1 + 4\gamma).$$

$\square$

## C   Proof of Corollary 1

*Proof.* It follows from Remark 1 that by invoking Theorem 2 with $\nu = \tau$ and

$$\gamma = a' \cdot \left(e^{2\tau} \cdot \frac{\rho\sqrt{d}\log D}{\sqrt{D}}\right),$$

for $a' > 1$, the following holds with high probability.

$$\left(1 - o_D(1)\right) \cdot \sum_{j\in\mathcal{N}_t} e^{o_j} \leq \frac{e^{o_i}}{q_i} \leq \left(1 + o_D(1)\right) \cdot \sum_{j\in\mathcal{N}_t} e^{o_j}. \tag{58}$$

Now, it is straightforward to verify that by combining (58) with (10) and (11), the following holds with high probability.

$$-o_D(1) \cdot \mathbf{1} \leq \mathbb{E}[\nabla_{\boldsymbol{\theta}}\mathcal{L}'] - \nabla_{\boldsymbol{\theta}}\mathcal{L} \leq \left(o_D(1) + o(1/m)\right) \cdot \mathbf{g} + \left(o_D(1) + o(1/m)\right) \cdot \mathbf{1}.$$

$\square$

## D   Toolbox

**Lemma 5.** *For a positive random variable $V$ such that $V \geq a > 0$, its expectation satisfies the following.*

$$\frac{1}{\mathbb{E}[V]} \leq \mathbb{E}\left[\frac{1}{V}\right] \leq \frac{1}{\mathbb{E}[V]} + \frac{\text{Var}(V)}{a^3}. \tag{59}$$

*and*

$$\frac{1}{\mathbb{E}[V]} \leq \mathbb{E}\left[\frac{1}{V}\right] \leq \frac{1}{\mathbb{E}[V]} + \frac{\text{Var}(V)}{\mathbb{E}[V]^3} + \frac{\mathbb{E}\big|V - \mathbb{E}[V]\big|^3}{a^4}. \tag{60}$$

*Proof.* It follows from the Jensen's inequality that

$$\mathbb{E}\left[\frac{1}{V}\right] \geq \frac{1}{\mathbb{E}[V]}. \tag{61}$$

For the upper bound in (59), note that the first order Taylor series expansion of the function $f(x) = \frac{1}{x}$ around $x = \mathbb{E}[V]$ gives us the following.

$$\frac{1}{x} = \frac{1}{\mathbb{E}[V]} - \frac{x - \mathbb{E}[V]}{\mathbb{E}[V]^2} + \frac{(x - \mathbb{E}[V])^2}{\xi^3}, \tag{62}$$

where $\xi$ is constant that falls between $x$ and $\mathbb{E}[V]$. This gives us that

$$\mathbb{E}\left[\frac{1}{V}\right] \leq \frac{1}{\mathbb{E}[V]} - \frac{\mathbb{E}\left(V - \mathbb{E}[V]\right)}{\mathbb{E}[V]^2} + \mathbb{E}\left[\frac{(V - \mathbb{E}[V])^2}{(\min\{V, \mathbb{E}[V]\})^3}\right]$$

$$= \frac{1}{\mathbb{E}[V]} + \mathbb{E}\left[\frac{(V - \mathbb{E}[V])^2}{(\min\{V, \mathbb{E}[V]\})^3}\right]$$

$$\overset{(i)}{\leq} \frac{1}{\mathbb{E}[V]} + \frac{\mathbb{E}\left[V - \mathbb{E}[V]\right]^2}{a^3}$$

$$= \frac{1}{\mathbb{E}[V]} + \frac{\text{var}(V)}{a^3}, \tag{63}$$

where $(i)$ follows from the assumption that $V \geq a > 0$.

For the upper bound in (60), let's consider the second order Taylor series expansion for the function $f(x) = \frac{1}{x}$ around $\mathbb{E}[V]$.

$$\frac{1}{x} = \frac{1}{\mathbb{E}[V]} - \frac{x - \mathbb{E}[V]}{\mathbb{E}[V]^2} + \frac{(x - \mathbb{E}[V])^2}{\mathbb{E}[V]^3} - \frac{(x - \mathbb{E}[V])^3}{\chi^4}, \tag{64}$$

where $\chi$ is a constant between $x$ and $\mathbb{E}[V]$. Note that the final term in the right hand side of (64) can be bounded as

$$\frac{(x - \mathbb{E}[V])^3}{\chi^4} \geq \frac{(x - \mathbb{E}[V])^3}{x^4}. \tag{65}$$

Combining (59) and (65) give us that

$$\frac{1}{x} \leq \frac{1}{\mathbb{E}[V]} - \frac{x - \mathbb{E}[V]}{\mathbb{E}[V]^2} + \frac{(x - \mathbb{E}[V])^2}{\mathbb{E}[V]^3} - \frac{(x - \mathbb{E}[V])^3}{x^4}. \tag{66}$$

It follows from (66) that

$$\mathbb{E}\left[\frac{1}{V}\right] \leq \frac{1}{\mathbb{E}[V]} + \frac{\text{Var}(V)}{\mathbb{E}[V]^3} + \mathbb{E}\left[\frac{\left|V - \mathbb{E}[V]\right|^3}{V^4}\right]$$

$$\leq \frac{1}{\mathbb{E}[V]} + \frac{\text{Var}(V)}{\mathbb{E}[V]^3} + \frac{\mathbb{E}\left[\left|V - \mathbb{E}[V]\right|^3\right]}{a^4}. \tag{67}$$

$\square$