[Reviews · NeurIPS 2019]

Reviewer 1



The presented method is an extension, using Random Fourier Features instead of the approximation used in [12], for kernel-based sampling on the evaluation of the gradients for SoftMax layer. It is not clear, and it is not studied at all, how the assumption of bounded gradients affects the empirical evidence. Same as above for the normalization of the class embeddings and input embeddings. Moreover, it is argued by the reviewer that such a process does not correspond to a widely SoftMax use. ----- After reading the rebuttal by the authors, I believe that this submission is stronger, however comments 1 and 2 are not addressed. I tend to keep my scores.

Reviewer 2



----- After rebuttal I have read all the reviews and believe the authors have done a good job addressing all the raised points. As a result, I will retain my scores and recommend this paper for acceptance. I kindly ask the authors to incorporate all the promised changes to the camera ready version. ----- The paper considers learning with conditional exponential family models and log-likelihood loss in cases when there is a very large number of classes. In such problems, it becomes expensive to evaluate the log-partition function for each instance from training sample. The main idea is to approximate the log-partition function by sampling a small number of scores corresponding to negative labels (different from the label assigned to a training sample). The model is given in Eq. (1), where the score for the i-th class is given by the inner product between a representation of an instance h and a parameter vector c_i representing the class. For a labeled example (x, y), the main idea is to sample a small number of negative labels from some distribution q defined on the label space and approximate the log-partition of the exponential family model using the target label y and the sampled negative labels (see Eq. 5). This reduces the computational cost of computing the log-partition from O(nd) to O(md), where n is the total number of categorical labels and m is the number of sampled negative labels. The key requirement on the sampling scheme is to show that the gradient estimate is unbiased (see Eq. 7). The main contribution is a theoretical result that provides a characterization for the bias of the gradient for a generic sampling distribution q (Theorem 1). This is a non-trivial result (with rather technical proofs) providing upper and lower bounds on the bias of the gradient. The discussion that follows shows that for label sampling strategy that is proportional to label scores, the bias tends to concentrate with rate o(1/m). This is a nice follow up discussion for a rather cumbersome formulation of the bound. The authors then proceed to define a kernel based model and observe that for this class of models and normalized (e.g., unit) vector representations of instances and class parameters, the label score can be represented with Euclidean distance between finite dimensional vectors || h - c_i || (see Eq. 13). Then, the score corresponding to an instance and a label can be factorized using random Fourier features and the proposed approximation of log-partition can be employed. The remainder of the paper discusses the interplay between kernel bandwidth parameter and scaling of the spectral measure used for sampling random Fourier features (Theorem 2). At this point, an assumption on the representation of instances and class parameters is introduced implicitly and deserves a discussion. Is this assumption really required and does the presentation require an excursion to kernels (there is an evident break between Sections 2 and 3)? The experiments illustrate the effectiveness of the approximation scheme using different datasets with the number of classes in range 10k -- 500k. This is a satisfactory number of classes and a good choice of benchmark datasets. The authors cover different settings and provide insights for normalized vs unnormalized representations of instances and classes, performance on extreme classification tasks, effects of kernel bandwidth and dimension, as well as some baselines. Summary The paper perhaps lacks a discussion on approaches that can avoid estimating the log-partition function such as minimization of Fisher divergence also known as score matching [1]. It would also be interesting to compare the proposed approximation scheme to this approach as a proper baseline because all the used baselines are based on a similar idea and rely on some distribution to sample a small number of negative labels (if I understood correctly). Additionally, an approximation scheme for log-partition function was considered in [2, Section 2.3] and it might be worthwhile to discuss it. Comments - Overall, I feel that the main part of the paper is relatively well-written (with some possible improvements to better link the sections). The theoretical result is significant and contributes towards understanding of the sampling strategies for approximation of the log-partition function. - It feels that there is a skip between Sections 2 and 3 and kernel-based exponential family models are not motivated appropriately. After the first reading, I was not really sure why random Fourier features were needed at all when one could have just as well defined an exponential family model with cosine features or any other suitable family of feature-functions. - The notation is not very clear, especially in the appendix. There are numerous expectation operators for which it is not clear with what variables they are coupled. It might be useful to add random variables and sampling distribution to the index of the operator. - In Eq. (5), I was not clear for the motivation of m until checking the appendix and reading the whole paper. It would be good to provide a sentence or two on the choice of scores o'. - Related work can also be improved and relevance of the work in the context of extreme classification could be stressed much more. References: [1] A. Hyvarinen (JMLR 2005). Estimation of non-normalized statistical models by score matching. [2] S. Vembu, T. Gartner, M. Boley (UAI 2009). Probabilistic Structured Predictors.

Reviewer 3



Extreme multiclass classification is an important and challenging problem: the naive approach to training involves calculating the (gradient of the) softmax loss function, which scales as the number of classes. One approach to make training more efficient is sampled softmax; however, bias is introduced from the sampling distribution differing from the softmax distribution. The authors propose a sampling method with reduced bias, based on random Fourier features (RF-softmax). They prove a bound for the bias of a sampling method in terms of the multiplicative factor difference from the softmax distribution, and bound the multiplicative factor difference for RF-softmax. They show an improvement in efficiency and accuracy over other sampling methods, on benchmark datasets such as Penn Treebank and Delicious-200k. The result is of practical importance, and the writing and organization are clear. The comparisons are thorough and well-documented: the authors document the effects of varying the dimension $D$ and temperature parameters $\nu$ and $\tau$. Suggestions and questions (in decreasing order of importance): + What do you get by applying Theorem 1 to RF-softmax? It would be nice to show as a corollary to Theorem 1 and 2 what the bounds are for RF-softmax. Are the bounds purely of theoretical interest, or do they give some guidance in selecting the parameters? + Theorem 2 and Remark 2: Where is the dependence on $D$ coming from in $o_D(1)$? I don't see a dependence on $D$ in (17). (Mention how $\gamma_2$ depends on $\gamma_1$.) + What happens if the inner product $\phi(c_i)^T\phi(h)$ is negative? + How does RF-softmax compare to hierarchical softmax? It appears that hierarchical softmax does not suffer from the shortcomings of negative sampling, as one can compute an unbiased gradient in $O(d \log n)$ time. What are the advantages of RF-softmax over hierarchical softmax? + $q_j$ should provide a tight uniform multiplicative approximation of $e^{o_j}$." How tight is the bound/how confident are you that this is the right bound? + It would be good to show the calculation that verifies (15). Minor edits: + (line 48) iterations -> iteration + (50) [brace] -> [parenthesis] + (55) compare -> compared + (130) based *on* + (144) multiplication -> multiplicative + (397) definition of $S_l$ should have indices $s_j$, not $j$. --- Reply: Thanks to the authors for addressing the points raised by the reviewers. I hope that the answers to the questions will be made clear, and that the theorem statement for RFF will be included in the final version of the paper. I continue to recommend the paper for acceptance.

[Author Response · NeurIPS 2019]

We thank the reviewers for their detailed and constructive feedback.

**R1:** *It is not clear... how the assumption of bounded gradients...:* We would like to point out that the assumption of
bounded gradient is only employed in the theoretical analysis of the bias. Such assumptions are quite common in the
analysis of ML algorithms (see e.g., [Hazan and Kale, 2014]). As mentioned in our submission, in many settings, this
holds because of the clipping of the gradients (see e.g., [Goodfellow, Bengio and Courville, 2016]).

*Same as above for the normalization of the class embeddings and input embeddings...:* As discussed in lines 167-174
of our submission, it is correct that we assume the embeddings to be normalized for RF-softmax, however, such
normalization is widely used in practice (e.g., see references [26], [27], and [28]). Furthermore, we have empirically
shown (on both NLP and extreme classification datasets) that with proper setting of $\tau$, the normalized embeddings do
not degrade the final performance of the model.

**R2:** *It feels that there is a skip between Sections 2 and 3 ...:* We thank the reviewer for the constructive comment. We
will enhance the presentation by better motivating the kernel-based sampling to approximate exponential families (i.e.,
RF-softmax method and its analysis) and smoothing the transition from section 2 to section 3. We have motivated the
use of random fourier features by comparing its performance against two most natural candidates in Table 1. We will
improve the relevant discussion in the final version.

*The paper perhaps lacks a discussion on approaches ... such as minimization of Fisher divergence ... an approximation*
*scheme for log-partition function was considered in [2, Section 2.3]...:* We thank the reviewer for the suggestion.
In [Hyvarinen 2005], the partition function $Z$ is just the function of model parameter and thus disappears in scoring
function. However, in our case, the partition function depends on the input $h$ (which changes during the training).
Therefore, while calculating the score function (taking derivative of $Z$ with respect to $(h, c)$), the partition function has
a non-trivial contribution. As for [Vembu et al., 2009], given ways to generate uniform samples for the set of classes,
they propose a MCMC approach to sample a class with a distribution that is close to full softmax distribution. Such
methods do not come with precise sample/computational complexity guarantees. We will include a discussion in the
final version.

*The notation is not very clear, especially in the appendix...:* We will highlight the distribution/random variables with
respect to which we take the expectations. Also, we will try to eliminate any inconsistency/ambiguity regarding the
notation elsewhere in the paper.

*In Eq. (5), I was not clear for the motivation of $m$ until checking the appendix and reading the whole paper...:* Adjusting
the logits for negative classes using their expected number of occurrence [Bengio Senecal, 2008] is critical for the
unbiasedness of sampled softmax loss, e.g., it ensures that $Z'$ is an unbiased estimator of $Z$. We will add a comment to
clarify the process of adjusting the logits in (5).

*Related work can also be improved and relevance of the work in the context of extreme classification...:* We will
highlight relevant papers in extreme classification literature in the final version.

**R3:** *What do you get by applying Theorem 1 to RF-softmax ...:* As pointed out in the discussion following Theorem 1,
the result provides a guidance for selecting a sampling distribution with low bias (by highlighting the requirement of
tight multiplicative approximation). We will combine Theorem 1 and 2 (at least in the setting with large D) in the final
version to obtain the bounds for RFF.

*Theorem 2 and Remark 2: Where is the dependence on D coming from in $o_D(1)$? ...:* Thanks for the comment. We
will include a comment on how $\gamma_2$ depends on $\gamma_1$. As it's clear from the proof of Theorem 2 and the statement of
Remark 2, one can choose $\gamma_1 = const\sqrt{(d \log D)/D}$. Now $\gamma_1$ (and thus $\gamma_2$) scales as $o_D(1)$ (while keeping other
parameters fixed).

*What happens if the inner product $\phi(c_i)^T\phi(h)$ is negative?:* With normalized embeddings, for finite $\nu$, $e^{(\nu h^T c_i)}$ is
strictly positive. Therefore, for reasonably large $D$, with high probability, $\phi(c_i)^T\phi(h)$ should be non-negative. In
general, one can replace $\phi(c_i)^T\phi(h)$ with $max(0, \phi(c_i)^T\phi(h))$ (with minor modifications in the proofs).

*How does RF-softmax compare to hierarchical softmax...:* As discussed in [Balnc Rendle, 2018] and references therein,
in many tasks, the final solution of hierarchical softmax is worse than those of both full-softmax and sampled-softmax.

*"$q_j$ should provide a tight uniform multiplicative approximation of $e^{o_j}$." How tight is the bound...:* We believe that the
bounds presented in Theorem 1 are fairly tight. In particular, these bounds recover the unbiasedness of the gradient for
full softmax distribution.

*It would be good to show the calculation that verifies (15):* (15) follows from the existing literature on RFF. We plan to
include a citation to [Yu et al. 2016, Lemma 1].

We will fix all the typos pointed out by the reviewers and eliminate other remaining typos in the final version.

[Meta-Review · NeurIPS 2019]

The reviewers all considered the author feedback and discussed the paper thoroughly. Most concerns could thereby be clarified but not all. Overall, I consider the contribution sufficiently interesting and valuable to justify accepting the paper as a poster.